



# Spatio-temporal assessment of the PCB sediment contamination in the four main French River Basins (1945-2018)

André-Marie Dendievel[1], Brice Mourier[1], Alexandra Coynel[2], Olivier Evrard[3], Pierre Labadie[2], Sophie Ayrault[3], Maxime Debret[4], Florence Koltalo[5], Yoann Copard[4], Quentin Faivre[6], Thomas Gardes[4], Sophia Vauclin[1], Hélène Budzinski[2], Cécile Grosbois[6], Thierry Winiarski[1] & Marc Desmet[6]

[1]Univ. Lyon, ENTPE, UMR CNRS 5023 LEHNA-IPE, Vaulx-en-Velin, 69120, France
[2]Université de Bordeaux, UMR CNRS 5805 EPOC, Bordeaux, 33615, France
[3]Institut Pierre Simon Laplace, UMR CNRS 8212 LSCE, Gif-sur-Yvette, 91191, France
[4]Université de Rouen Normandie, UMR CNRS 6143 M2C, Mont-Saint-Aignan, 76821, France
[5] Université de Rouen Normandie, UMR CNRS 6014 COBRA, Mont-Saint-Aignan, 76821, France
[6]Université de Tours, EA 6293 GEHCO, Tours, 37200, France

*Correspondence to*: André-Marie Dendievel (andre-marie.dendievel@entpe.fr) and Brice Mourier (brice.mourier@entpe.fr)

## Short Summary

A dataset of PCBi data from sediment cores, bed and flood deposits, suspended particulate matters and dredged sediments on the main French rivers (source–estuary transects; 1945–2018) is compared with socio-hydrological drivers. PCBi concentrations increased from 1945 to the 1990s, due to urban or industrial emissions. It decreased with the implementation of regulation. Computed specific PCBi fluxes confirmed the dispersion of heavily polluted sediments by the French Rivers in European Seas.

## Abstract

Environmental pollution by Polychlorinated Biphenyls (PCBs) is a key concern about river quality because of their low degradation rates leading to their accumulation in sediments or living organisms. This original interdisciplinary work was conducted at a large scale along the four main French rivers (Seine, Rhône, Loire and Garonne Rivers), which flow into major European seas. We completed a dataset based on sediment analyses provided by monitoring agencies, port authorities and research teams on different solid matrices (cores, bed and flood sediments, suspended particulate matters, dredged sediments). This dataset (n=1416) focused on the seven indicator PCBs and their sum ($\Sigma$PCBi) from 1945 to 2018. Special effort was put on the quality control to provide robust spatio-temporal information. Taking into account hydrological and human drivers, we outlined two main pollution trends: (1) from 1945 to 1975, a quick increase of $\Sigma$PCBi (up to 4 mg.kg$^{-1}$ dry weight, dw) and a sharp decrease in the 1980s on the Seine and Loire Rivers; (2) increasing but moderate $\Sigma$PCBi levels (50 to 150 µg.kg$^{-1}$ dw) followed by a decline after the 1990s on the Rhône and Garonne Rivers. In addition to these patterns, PCBs emissions from urban and industrial areas or accidental events were significant on each river. Finally, when calculating specific fluxes, the Rhône River exhibited the biggest $\Sigma$PCBi load (up to 12 µg.m$^{-2}$.year$^{-1}$ in 1977-1987), at least 25 % higher than those of the Seine and Loire Rivers, while the Garonne River showed very low fluxes. French specific $\Sigma$PCBi fluxes are two orders of magnitude lower than those found in American or Asian Rivers. In Europe, we confirmed that the Rhone and Seine Rivers provide a major supply of PCBi to the Western Mediterranean and the English Channel, respectively. The dataset is available at https://doi.pangaea.de/10.1594/PANGAEA.904277 (Dendievel et al., 2019).

## Keywords

Polychlorinated Biphenyls, Persistent Organic Pollutant, Rivers, Sediment, Cores, Monitoring, Pollution trajectories, Specific fluxes.



## 1 Introduction

Environmental pollution of river sediments due to Polychlorinated Biphenyls (PCBs) has become a concern as early as in the 1970s (Dennis, 1976; Müller, 1986). Indeed, PCBs were then widely used as heat transfer fluids and insulating fluids for transformers and capacitors, while they had been utilized as wood, paper, plastic or ink additive since 1930 worldwide (Breivik et al., 2002a; De Voogt and Brinkman, 1997). Due to their high toxicity and their long persistence in the environment, the use of PCBs was banned in the USA (Toxic Substances Control Act of 1976) as well as in the Organisation for Economic Cooperation and Development (OECD) member countries. In France, the use of PCBs was progressively restricted to closed systems, i.e. capacitors and transformers, by the order of 08/07/1975. Then, it was entirely prohibited by decree in 1987 (n° 87-59). Old devices using PCBs are currently being dismantled after European Guidelines (2001-63 and 2013-301 national decrees). To support these provisions, a global survey of PCBs in water, sediment, fish and bryophytes has started in the 1990s in France on behalf of the Survey and Control Network (RCS), jointly managed by the Water Basin Agencies (WBA) and the Regional Directorate for Environment, Development and Housing (DREAL). In charge of the local sampling and analyses, WBA focused on seven PCB congeners (PCB-28, -52, -101, -118, -138, -153, -180) and their sum, referred to as $\sum$PCBi in the remainder of the text. These PCB indicators are generally found in high concentrations in the environment (e.g. sediment and biota), as well as in human food (IARC, 2016). Such provisions, taken at the worldwide scale, have led to a significant reduction of the quantity of PCBs found in the environment (Breivik et al., 2007; Wania and Su, 2004).

However, PCBs stored in the atmosphere, in soils and in hydrosystems still remain a major issue as revealed by recent studies on marine mammals and ice sheets contamination (Desforges et al., 2018; Hauptmann et al., 2017). Moreover, high PCB levels found in estuarine or riverine fauna – mussels, oysters, eels or fishes – are directly attributed to upstream sediment contamination (Blanchet-Letrouvé et al., 2014; Olenycz et al., 2015). In Europe and around the Mediterranean Basin, several studies compared sediment and biota concentrations (Adda River, Italy: Bettinetti et al., 2016; Seine River, France: Chevreuil et al., 2009; Rhône River, France: Lopes et al., 2012; Thames River, UK: Lu et al., 2017; Elbe River, Germany: Schäfer et al., 2015; Nile River, Egypt: Yahia and Elsharkawy, 2014). However, in most of these cases, understanding the PCB contamination fluxes remains complex since PCBs are stored in sediments from oxbow lakes, dams, soils, dumping areas along rivers and coasts. Furthermore, this contaminated material is known to move through the system as suspended particulate matters (SPM) and could be submitted to successive deposition and remobilization stages (floods, flushing, tidal effect, etc.). Diverse regulatory assessment and quality scale across countries (sampling frequencies and stations, analytical methods, limits of quantification, studied PCB-congeners) complicate the estimation of PCB stocks in rivers at global scales. As for other pollutants, a more integrative research framework combining long-term data (i.e. sediment cores) and continuous/frequent monitoring is needed to capture the spatial and temporal variability of the contamination by PCBs and to identify the drivers of this variability (Meybeck et al., 2018).

The current research aims at comparing PCB contamination in sediments along four major rivers with hydrological and human drivers over the period 1945–2018. In order to evaluate the magnitude of PCB fluxes transiting from the rivers to the seas at a nationwide scale, we focused on the French rivers that flow into several major seas of the Northern Hemisphere (Seine River: English Channel and North Sea; Loire and Garonne Rivers: Atlantic Ocean; Rhône River: Mediterranean Sea). These rivers are also known to have been strongly modified by anthropogenic activities since the 19th century to facilitate the fluvial transport and, later on, for hydroelectricity and nuclear power purposes through the construction of dams, weirs and diversion canals (Tricart and Bravard, 1991; Parrot, 2015; Lestel et al., 2019). The occurrence of lateral structures to narrow and straighten the river stems is also typical (Lower Loire: groynes called "épis de Loire" – Barraud et al., 2013; Rhône River: "Girardon" infrastructures; Tricart and Bravard, 1991; Lower Garonne: Lalanne-Berdouticq, 1989; etc.) and induced the storage of fine sediment with variable contents of organic pollutants in the floodplains (Vauclin et al.,2019).

Because validated historical data covering a wide range of locations along each river were required to approach spatio-temporal contamination trajectories, we compiled a dataset focusing on the sum of the seven indicator PCB ($\sum$PCBi= PCB-28 + PCB-52 + PCB-101 + PCB-118 + PCB-138 + PCB-153 + PCB-180). $\sum$PCBi data were acquired on sediment cores, fresh bed and flood sediments and SPM or during dredging operation on the French rivers (Dendievel et al., 2019). We controlled data quality, spatial and temporal coverage to support our interpretation of the dataset and to maximize its robustness. This research also investigated how spatio-temporal contamination trends may be related to population, riverside land use (urban and industrial areas) and accidental PCB releases along each river. Finally, specific fluxes of PCBi ($\mu$g.m$^{-2}$.year$^{-1}$) were calculated upstream of estuarine areas. The results were compared to worldwide data in order to propose a more integrative estimation of the mass contribution of the main French Rivers to the global PCB pollution transferred to the sea.



## 2 Methodology

### 2.1 Data collection on the studied rivers

100 Sediment contamination was assessed by collecting PCBi analysis results and associated ancillary data (total organic carbon content, grain size) at gauging stations located on the main stream and monitored by regulatory water agencies or scientific teams. Significant effort was made to collect high quality information from multiple sources (harbour and navigation authorities, WBA, DREAL, public research labs, etc.) at a national scale during the 1945-2018 period (initial total number of samples ~2300). The data came from four main sediment matrices:

105     - (1) Bed and flood sediments from both upper and lower catchment areas are assumed to be deposited sediments (monitoring: DREAL and WBA or ROCCHSED program of the IFREMER, respectively). Sampling was performed by field operators since 1991 – 1995 on 135 stations distributed along the main rivers (see Fig. 1 and supplement table 1). Sampling frequencies ranged from once or twice a year to a unique sample in 30 years. All data – including sampling locations, dates, results and techniques if 110     mentioned – were aggregated from 3 open databases: (a) "naiades" supplied by WBA (http://www.naiades.eaufrance.fr; accessed October 15, 2018), (b) the National Actions Plan on PCB (http://www.pollutions.eaufrance.fr/pcb/resultats.html; accessed January 15, 2019), (c) "surval" powered by the IFREMER (https://wwz.ifremer.fr/surval; accessed January 15, 2019). Academic studies focusing on flood deposits were also included (e.g., Lauzent, 2017).

115     - (2) Suspended Particulate Matters (SPM) are considered as mobile sediments. SPM concentrations were monitored in the Rhône River by the OSR (Rhône Sediment Observatory) since 2011 at two stations: Jons and Arles, which are located upstream of Lyon and just upstream of the delta, respectively (https://bdoh.irstea.fr/observatoire-des-sediments-du-Rhône; accessed May 19, 2019). DREAL and WBA were in charge of the SPM monitoring since 1993 on the Garonne River (three stations: upstream = Verdun-120     sur-Garonne; middle Garonne = before the Lot confluence; downstream = Cadillac) and on the Loire River (four stations: upstream = Veauchette; downstream = La Possonnière, Montjean and Ste-Luce). We did not include SPM data from the Seine River because results were only available for "bulk water" (g.L$^{-1}$) rather than for sediment. SPM were measured monthly to quarterly, sometimes bi-monthly, on the Garonne, Loire and Rhône Rivers.

125     - (3) Mobile dredged sediments were collected and analysed by harbour and navigation authorities. ∑PCBi data were provided by the Port of Rouen for the lower and estuarine sections of the Seine River with 5 stations monitored since 1992, and by the CNR (Rhône National Company) on the Rhône River with 81 sampling locations mostly distributed next to hydraulic infrastructures or at river confluences.

    - (4) Sediment cores deposited in reservoirs, oxbow lakes or channel banks were extracted by the research 130     teams from the INTERPOL consortium. Among French rivers, the Rhône was intensively investigated with 13 cores analysed for PCB contents along the whole river (Fig. 1). These studies provided vertical profiles of historical contamination on each river section since 1945 at least (Desmet et al., 2012; Mourier et al., 2014). On the other rivers, most analyses focused on lower sections to estimate the pollution trends at the river mouth: 3 cores were extracted downstream of Paris on the Seine River, providing records since 1945 135     (Boust et al., 2012; Lorgeoux et al., 2016; Vrel et al., 2013), 1 core was extracted from the Garonne River, downstream of Bordeaux, dating back to 1954 (Budzinski, Labadie and Coynel, pers. com.; Morelli et al., 2016; Adapt'Eau project) and 1 core from the Loire River, near Nantes (Desmet, personal communication; Metorg project) covering the period since the late 1970s. On the latter river, a second core was also analysed for Persistent Organic Pollutants directly downstream of the industrial basin of Saint-Etienne (Fig. 1; 140     Bertrand et al., 2015; Grosbois, personal communication; Metorg project).

### 2.2 Analysis and quality control

PCB-congener analyses in riverine sediments were performed according to several methods depending on stakeholders and research labs. It generally followed four main steps. (1) Sampling was achieved by using Ekman 145 grabs for bed and flood deposits, fluvial decanters for SPM, and hydraulic excavator for dredging. Cores at onshore sites were collected with percussion corers (e.g. Cobra TT) while those underwater were sampled with piston corers (e.g. UWITEC). Samples were directly extracted in the laboratory and freeze-dried for conservation. Sieving was sometimes conducted when coarse sediment was collected (sieving to 2 mm or 63 µm). (2) According to our review of the methods, the extraction was usually achieved with Soxhlet or microwave-assisted before (3) a purification



step via adsorption chromatography with sorbents such as silica, aluminium oxide, Florisil and activated carbon. (4) PCB-congeners were quantified by gas chromatography coupled with mass spectrometry (GC-MS) in selected ion monitoring (SIM) mode (for details, please refer to the related papers and sources cited in Table 1).

The quality control of available monitoring data was one of our major challenges given the large variety of data collected. Indeed, PCBi results from bed and flood deposits, SPM or dredged sediment included missing values,
outliers and variable limits of detection (LODs) or quantification (LOQs) that have changed over time. According to the employed analytical methods, LOQs ranged between 1 and 140 µg.kg$^{-1}$ dw for ∑PCBi (i.e. 0.1 to 20 µg.kg$^{-1}$ dw by congener). Moreover, among the seven PCBi-congeners, the low chlorinated compounds (PCB-23, PCB-52, PCB-101 or PCB-118) were often not quantified due to higher LOQs. Accordingly, to maximize the robustness of the analysis, the dataset includes results having all seven PCB congeners > LOQs. In addition, we corrected the
sample results where high-chlorinated PCB-congeners, namely PCB-138, PCB-153 and PCB-180, were > LOQs. For this correction, we calculated the ∑PCBi according to the following equation (1):

$$\sum PCBi = \frac{\sum PCBHC \times 100}{TM\ PCBHC}\ (1)$$

where ∑PCBHC is the sum of high-chlorinated PCB-congeners (PCBHC), i.e. PCB-138 + PCB-153 + PCB-180 (µg.kg$^{-1}$ dw) and TM PCBHC is the mean percentage of the three PCBHC in well quantified samples on each river
(TM PCBHC$_{Garonne}$= 63±11 %; TM PCBHC$_{Loire}$= 68±12 %; TM PCBHC$_{Rhône}$= 60±12 %; TM PCBHC$_{Seine}$= 55±10 %).

As showed in Table 1, this step increased the proportion of validated data from 6 % to ca. 18 % for the Garonne River (48 samples), from 21 % to 36 % for the Loire River (147 samples), from 32 % to 44 % for the Rhône River (294 samples) and from 58 % to 76 % for the Seine River (343 samples). Table 1 also presents total organic carbon
(TOC) and fine fraction content (% of clays and silts) of the investigated sediments. This additional data could not be used for normalization because it was not systematically available for each sediment matrix or sample and also because each basin has specific geochemical background signatures.

## 2.3 River system characterisation

Longitudinal river profiles, catchment surfaces, population in the drainage area and distances to fluvial estuary for each sampling station were collected by using IGN tools and services (French National Institute of Geographic and Forest Information): Géoportail (https://www.geoportail.gouv.fr; accessed November 30, 2018), BD ALTI®, GEOFLA® and BD Carthage® (http://professionnels.ign.fr; accessed November 30, 2018). Annual average water discharges (m$^3$.s$^{-1}$) were computed according to the national databank of hydrological information ("Banque
Hydro"; http://www.hydro.eaufrance.fr; accessed May 19, 2019). Other fluvial corridor characteristics were collected from the European Corine Land Cover (2012) for urban and industrial areas (v.20b2, https://land.copernicus.eu/pan-european/corine-land-cover; accessed May 19, 2019), and from *BASOL* (https://basol.developpement-durable.gouv.fr/home.htm; accessed February 11, 2019) and *BASIAS* (http://www.georisques.gouv.fr/dossiers/basias/donnees#/; accessed May 4, 2019) databases for the location of the
PCB polluted sites. For the latter, we considered: (1) confirmed pollution after accidental spillage from electrical transformers or contaminated sludge and (2) suspected contamination due to past activities using PCBs such as ink or plasticizer manufacturing and storage (see supplement table 2). The *QGIS* software (v.2.18.28) was used to merge and intersect all the data in a buffer zone of 1 km on both sides of each river.

## 2.4 Time series analysis and specific flow rates


The 1416 validated ∑PCBi data points (Dendievel et al., 2019) were analysed as a whole and represented according to temporal trends since 1945 in *R* (v3.5.1, R Core Team, 2018) with the package "ggplot2" (v.3.1.1, Wickham et al., 2019). We used general additive models (gam) within the "stat_smooth" function to draw robust non-parametric models, little influenced by outliers and not truncated at the end. The basis dimension k was adjusted by using the
"gam.check" function, available in the "mgcv" package (Wood, 2019). Chronological control was based on the date of sampling for regulatory monitoring data (bed and flood sediments, dredging and SPM). For core data, we used published chronological models (for instance, see Desmet et al., 2012; Lorgeoux et al., 2016; Mourier et al., 2014). To discuss more accurately spatial and temporal distribution, we proposed a boxplot analysis according to river sections (upper, middle and lower sections) and based on three main time windows defined by hierarchical





clustering ("Chclust" function in *R*; Juggins, 2013): < 1997, 1997–2007, 2007–2017. Then, we used the annual average of $\sum PCBi$ concentrations in sediments from lower river sections to calculate the $\sum PCBi$ load (t.year$^{-1}$). This calculation was based on annual water discharge (m$^3$.s$^{-1}$) and SPM concentrations (mg.L$^{-1}$) relationships since 1977 at the main gauging stations located upstream of tidal influence zones (Fig. 1; for specific equations see: Coynel et al., 2004; GIP Seine Aval, 2008; Moatar and Dupont, 2016; Poulier et al., 2019). Finally, we normalized

this load to the drainage area at each gauging station to get specific flow rates of $\sum PCBi$ (µg.m$^{-2}$.year$^{-1}$).

## 3 Results

### 3.1 Comparability of the results in the different sediment matrices

The $\sum PCBi$ distributions in each sedimentary matrix were distinguished following two groups characterized by similar hydro-sedimentary settings: (1) "deposited sediments" including bed or flood deposits and core sediments,

(2) "mobile particles" including transiting sediments such as SPM and dredged sediments. $\sum PCBi$ distributions between both groups were compared for each river to determine whether these datasets were statistically comparable and therefore suitable for supporting the discussion of the spatio-temporal contamination trends in the different rivers. The comparison was performed on a common time period and spatial area for which both groups of data were available (Figs. 1 and 2). For instance, for the Rhône River, we selected the $\sum PCBi$ data covering the

period 2011–2018 on the whole river, because SPM data (mobile particles) were only available after 2011. In a similar way, we compared deposited and SPM data on the estuarine Seine (lower 50 km) and on the Upper Loire (1012 to 750 km) sections since 1992 and 1991 respectively, because analyses on mobile sediments were available earlier than for the Rhône River. Regarding the Garonne River, we compiled all the available data on the lower section (last 100 km), including SPM, cores or bed and flood sediments since 1954. Graphical and related Wilcoxon

paired tests showed that $\sum PCBi$ were not significantly different between matrices (p > 0.05) and could therefore be compared to conduct trend analyses in a robust way (Fig. 2).

### 3.2 Longitudinal river profiles, dams and water discharge

Physical settings and river modifications for navigation, flood control or electricity generation could be considered

as major factors influencing the sedimentation and the transport of polluted sediment. Accordingly, the studied rivers were divided into two groups based on the degree of water engineering disturbances and on flow rate trends (Table 2).

The first group is characterised by step-like longitudinal profiles due to heavy engineering caused by the installation of multiple hydroelectric plants and dams, changing the natural geomorphological dynamics. It comprises the Seine

and the Rhône Rivers (Fig. 3-a and 3-e).

The Seine Valley is adapted for fluvial transport from and to Paris. It is equipped with 23 small dams and weirs currently managed by the Seine Navigation Service. The main hydraulic structure between Paris and Rouen is the Poses dam, which regulates the stream and stops tidal waters. Major tributaries flow into the Seine halfway between the source and the estuary, in urban areas around Paris Megacity where the junction with the Yonne, Marne and

Oise Rivers occurs (Fig. 3-a and 3-e). On the Lower and estuarine Seine, the Eure River is the main tributary (annual discharge Q= 26 m$^3$.s$^{-1}$). At the estuary, the Seine River has a current discharge of ca. 600 m$^3$.s$^{-1}$ (Table 2).

The Rhône River corridor was also substantially modified for navigation purposes and hydroelectricity production. Several hundreds of "Girardon infrastructures" (dykes and groynes systems) were built to reduce the stream width and secondary channels were partly disconnected. Nowadays, 19 dams are still managed by the CNR (Rhône

National Company). Major dams are located in the upper sections including the Génissiat Dam, which is one of the highest French dams (104 m). The Upper Rhône is mainly supplied by Alpine rivers. The other main tributaries come in the medium and lower sections, in densely urbanized areas including Lyon, with the Saône River confluence (Q= 410 m$^3$.s$^{-1}$), Valence and Avignon with the Isère and Durance confluences, respectively (Q= 333 and 180 m$^3$.s$^{-1}$). At the mouth (Rhône delta), the flow rate culminates at ca. 1700 m$^3$.s$^{-1}$ with a high solid transport

estimated from 4 to 8 Mt of sediments each year (Table 2).

The second group is composed by rivers having a smooth longitudinal profile, which are relatively close to an equilibrium profiles, i.e. the Garonne and Loire Rivers (Fig. 3-i and 3-m). These rivers could be considered as less modified than those of the first group although they are not devoid of hydraulic structures.

The Loire River is the longest French River (1006 km) and one of the main rivers flowing into the Atlantic Ocean





in Western Europe. Its main stem is equipped with three major dams at La Palisse, Grangent and Villerest, managed by EDF for hydroelectricity and flood protection. Other significant hydraulic infrastructures are located in Lower Loire where secondary channels were closed and groynes were built to increase the water level for navigation purposes (Fig. 3-i n°9). Middle and Lower Loire sections are also equipped with weirs and small reservoirs for nuclear power plants (Fig. 3-i). The location of the main confluences delineates the boundaries of the fluvial

sections: the Allier River ($Q= 140$ $m^3.s^{-1}$) is the downstream boundary of the Upper Loire; the Cher, Vienne and Maine Rivers represent the transition between Middle and Lower Loire sections ($Q=104$, $203$ and $127$ $m^3.s^{-1}$ respectively). The flow rate reaches ca. $870$ $m^3.s^{-1}$ in the Lower section.

In the same group, the Garonne River is mainly equipped with high dams on its upper section (Pyrenees foothills) such as Pont-du-Roi dam at the border between France and Spain (Fig. 3-m). One major structure modifies the

main stream on the middle section: the channel of the Golfech nuclear power plant. In the lower section, the Garonne width is restricted by slide structures increasing the water level for navigation. Flow rates show increases due to the contribution of successive tributaries from upper (Ariège River) to lower sections (Tarn and Lot Rivers) with a total ca. $650$ $m^3.s^{-1}$ at the Gironde estuary. Solid fluxes range from 0.9 to 3 $Mt.year^{-1}$ in relation with annual hydrological conditions – i.e. dry or humid years.


### 3.3 Land Use and Population

To support the comparison of $\sum PCBi$ data and to understand the spatio-temporal trends of contamination between the studied rivers, we also acquired population and land use data for each river corridor (Fig. 3; see also suppl. table 2).

The first group (Seine and Rhône Rivers) is highly populated halfway between the source and the estuary. This tipping point is also associated with the occurrence of major urban and industrial areas nearby the river. On the Seine River, a major increase in population and a concentration of industries occur in Paris Megacity, the largest urban centre in France ($10.6$ $M_{inhab}$; Fig. 3-b). Although other urban and industrial areas may be found, such as at Troyes ($130$ $K_{inhab}$), at the Yonne confluence (Upper Seine) or next to Rouen ($405$ $K_{inhab}$; Lower Seine), they remain

small compared to Paris (Fig. 3-b). In a similar way, the Rhône River is characterised by increasing population densities in a downstream direction with a demographic upsurge at Lyon (330 km before the sea). Near the Lyon Megacity ($2.3$ $M_{inhab}$), urban areas occupy up to 57 % of the river corridor and industrial areas cover 16 to 22 % of the surface area (Fig. 3-f), in particular in the so-called Chemical Corridor, south of Lyon.

The second group (Loire and Garonne Rivers) shows more gradual increasing population densities along the river

with step-like demographic curves due to the regular presence of cities and towns (Fig. 3-j and 3-n). On the Loire Basin, urban areas (25 to 50 %) and related industrial areas (ca. 5 %) follow one another, such as Nevers, Orléans, Tours or Angers (Fig. 3-j). However, two main historical industrial basins are also found: St Etienne Metropolis ($404$ $K_{inhab}$) in the Upper Loire (31 % urban and 19 % industries) and Nantes Valley next to the Loire estuary (20 % urban and 30 % industries; $640$ $K_{inhab}$). On the Garonne watershed, there are also two main cities and industrial

areas (Fig. 3-n): Toulouse Metropolis, 370-280 km from the Garonne estuary (32 % urban, 12 % industries; $760$ $K_{inhab}$), and Bordeaux Metropolis (32 % urban, 19 % industries; $780$ $K_{inhab}$), 50-25 km from the Garonne estuary.

### 3.4 PCB pollution along the studied rivers

Physical settings, land use and pollution site distribution are compared with the $\sum PCBi$ spatial patterns in sediments along the studied rivers in Fig. 3-c, 3-g, 3-k and 3-o. Overall, maximum values are recorded on the Lower Seine

River in the 1970s near Rouen (up to 5 $mg.kg^{-1}$ in the Darse des docks record). On the Rhône River, the highest $\sum PCBi$ concentrations are found in the Middle Rhône, downstream of Lyon in 1995–1996 (up to 2.4 $mg.kg^{-1}$). On the Loire Valley, $\sum PCBi$ peaks in the St Etienne Basin from 1966 to 2006 (upper section; up to 1.1 $mg.kg^{-1}$) and in the Nantes Valley between 1973 and 1989 (lower section; 0.6 to 1.2 $mg.kg^{-1}$) and also sporadically in 2003–2008 (up to 1.4 $mg.kg^{-1}$). $\sum PCBi$ in the Garonne sediments are generally low and the maximum concentrations are

recorded near the city of Toulouse in 1998 (ca. 145 $\mu g.kg^{-1}$).

The Seine River in particular shows an increasing $\sum PCBi$ trend in the downstream direction (Fig. 3-c). Rather low $\sum PCBi$ concentrations are measured upstream of Paris (median: $28\pm20$ $\mu g.kg^{-1}$). An increase is obvious from Paris (median: $103\pm79$ $\mu g.kg^{-1}$) to Rouen (median: $318\pm348$ $\mu g.kg^{-1}$). Part of this increase might be due to the Eure Valley inputs (140 km upstream of the estuary) which hosts important pharmaceutic and other industrial facilities

(Fisson et al., 2017; Gardes et al., submitted). Maximum $\sum PCBi$ concentrations ranged from 0.5 to ca. 5 $mg.kg^{-1}$. A decline is observed in the lower 80 km, between Rouen and Le Havre ports, where $\sum PCBi$ decreased to ca. $25\pm12$



μg.kg⁻¹ in estuarine zones. On the Rhône River, based on 13 historical cores along the river, we highlight an increasing ∑PCBi trend from the Upper Rhône (median: 15±10 μg.kg⁻¹) to the Middle Rhône section (median: 32±24 μg.kg⁻¹). In addition, samples collected in the Chemical Corridor and near the Gier confluence (Middle

Rhône section) show very high contamination levels (ca 2 mg.kg⁻¹). Then, the Lower Rhône shows a decreasing trend with a slight dilution of the PCB contamination (median: 24±18 μg.kg⁻¹).

The global distribution of ∑PCBi in the Loire is mainly driven by two major areas: the industrial basin of St Etienne - Roanne (Upper Loire; median = 153±101 μg.kg⁻¹) and the Nantes Valley (Lower Loire; up to 1.4 mg.kg⁻¹; Fig. 3-k). Between these two sectors, reduced ∑PCBi concentrations are recorded while the low density of observations

in the Middle Loire is not sufficient to conclude. The same observation applies to the whole Garonne River. Indeed, continental sections of the Garonne River really lack accurate monitoring or historical data about persistent organic pollutants to build a clear link between land-use and pollution levels (Fig. 3-o).

According to *BASIAS* and *BASOL* databases, PCB contaminated sites are reported along the rivers in Fig. 3-d, 3-h, 3-l, and 3-p. These databases provide valuable insights into the main polluted sites that should be cleaned up in

priority. PCB contaminated sites were divided into two categories according to confirmed events (leaks or fires from electrical capacitors or transformers) or to suspected PCB pollution events (open environment uses: wood, paper, plastics, inks, flammable fluids). High frequency of PCB-polluted sites is evidenced next to Paris and Rouen (Seine), Lyon (Rhône), Toulouse and Bordeaux (Garonne) conurbations. The spatial link between these sites and the location of major urban or industrial areas is clear because electrical power centres are specifically established

to supply these areas. On the Loire River, any relationship between pollution events and other settings remains unclear and numerous incidents took place along the river (Fig. 3-l). Such divergences are most certainly due to an incomplete listing or survey of pollution events (Callier and Koch-Mathian, 2010).

## 4 Discussion

### 4.1 Temporal trends of the PCB contamination since 1945 in the main French rivers

In addition to spatial trends, ∑PCBi temporal trends are discussed to provide a dual analysis of the evolution of the contaminant concentrations in sediments. We are able to describe environmental histories of rivers at different timescales: since 1945 for the Seine and Rhône Rivers, and since 1973 and 1954 for the Loire and Garonne Rivers, respectively.

Among the reconstructed trends, the Seine curve shows a close relationship with the theoretical production trend

(Figs. 4-a and 4-f). Four main steps are highlighted on the Seine River (Fig. 4-a): (1) ∑PCBi gradually increased from 1945 to 1970, with a plateau at the end of the 1950s known at both local and global scales (Breivik et al., 2002a; Lorgeoux et al., 2016). (2) The Seine curve reached a maximum in 1975 with ca. 1800 μg.kg⁻¹ of ∑PCBi. Then, (3) it sharply decreased to ca. 300 μg.kg⁻¹ in the late 1980s and (4) to ca. 100 μg.kg⁻¹ in the 2000s (Fig. 4-a). The median level of current ∑PCBi contamination remained above the lower effect level at which toxicity to

benthic-dwelling organisms are predicted to be unlikely (threshold effect concentration – TEC = 59.8 μg.kg⁻¹ according to MacDonald et al., 2000) or above the level below which dredging and relocation activities would be authorised by French authorities without further investigations (N1= 80 μg.kg⁻¹, GEODE). Figures 3-c to 3-d suggested that the origin of the PCB pollution is located in Paris or in Rouen urban and industrial areas, where long-term pollution is recorded, together with accidental contamination (more than 200 PCB polluted sites listed

in Paris according to *BASIAS* and *BASOL* databases). Inputs from the Eure Valley, just before Rouen, may also be considered (Fig. 3-c). The decline from steps (3) to (4) seems linked with the prohibition of production, sale and purchase of devices using more than 500 mg.kg⁻¹ of PCB (mainly electrical transformers) after 1987 and their disposal according to the 1996 European Directive (French decree of 2003; Fig. 4-f).

The Loire record is shorter although the high ∑PCBi concentrations found in sediments between 1973 and 1978

(1,200 μg.kg⁻¹) may reflect a similar situation (step 2). A quick decrease to 10 μg.kg⁻¹ after 2010 could be related to steps 3 and 4 (Fig. 4-b). These changes are likely linked with global emission trends observed at the Western European scales according to Breivik et al. (2002a, 2002b, 2007) (Fig. 4-e). Indeed, after a global increase of PCB production and emissions from 1930 to 1970 corresponding to step 1, a sharp drop occurred after 1973 (step 2), linked to national applications of global regulations (OECD) prohibiting the use of PCB in open environments (Fig.

4-f). The fast decrease of the PCB concentrations in sediments in the late 1970s in the Seine and Loire Basins (half-life $t_{1/2}$ in sediments ranges from 5 to 13 years respectively) suggests a signal linked to the global reduction of PCB emissions (Rosen and Van Metre, 2010). Moreover, PCBi half-life decay in the Seine and Loire sediments ($t_{1/2}$) are





also consistent with those measured on the Rhône River cores where $t_{1/2}$ are comprised between 2 and 13 years (Desmet et al., 2012).

For the Rhône River, our model suggests a smoother ∑PCBi trend (Fig. 4-c). Indeed, the few concentrations found in the 1950s are ca. 70 µg.kg$^{-1}$ and slightly increased to a plateau of ca. 80 µg.kg$^{-1}$ in the 1980s. Finally, a general decrease is observed after 1996 (Fig. 4-c). This model, based on a large number of data, confirms the first modelling attempt of Desmet et al. (2012) based on a lower number of observations (four cores). The long-term plateau effect is obviously linked to monitored samples with high ∑PCBi in the 1980s and 1990s. Maximum values ranged from

0.7 to 2.4 mg.kg$^{-1}$ in 1995–1996 (i.e. much higher than regulation levels). They originate from Givors and St-Vallier, two stations located immediately downstream of the Lyon's Chemical Corridor and at the mouth of the Gier industrial Valley known for intensive mining, great smelters, forges and factories since 1870 (Gay, 1996). Within Lyon (Fig. 3-h), accidental contaminations of alluvial groundwater in abandoned or decommissioning industrial and commercial sites are also recorded from the west (Vaise, 300 L of Pyralene spilled in 1995) to the

north-east (Vaulx-en-Velin, 4,000 L in 2008). In this context, the delayed reduction of PCBi levels in the Rhône Corridor could be a late effect of the national regulation with a time-lapse of about 20 years compared to the situation observed for the Seine and Loire Rivers. It could be due to accidental releases in the middle section, but it is also certainly linked with river management and hydrological settings. Indeed, the Rhône River is equipped with major dams and slide structures ("Girardon infrastructures") which can store contaminated sediments for a

while. In addition, the Rhône River has the highest flow rate in France (ca. 1,700 m$^3$.s$^{-1}$) which has likely diluted the pollution levels measured in sediments and its floods could produce massive remobilization of contaminated sediments.

On the Garonne River, a preliminary modelling of the ∑PCBi contamination in fluvial sediments can be applied carefully based on few data > LOQs (Table 1; Fig. 4–d). According to this first attempt, ∑PCBi contents increased

until 1980 – 1990 and then progressively decreased. Figure 4–d displays a curved shape although the lack of accurate monitoring data and the reduced number of sedimentary archives (only one sediment core) do not allow distinguishing local from long-distance pollution. In any case, median ∑PCBi concentrations in Garonne sediments vary from less than 20 µg.kg$^{-1}$ (1950s and 1960s, 2000s and 2010s) to ca. 70 µg.kg$^{-1}$ (1980s and 1990s). Low ∑PCBi concentrations could be due to high sedimentation rates upstream of the monitored sites; whereas the

highest concentrations could be partly related to the reworking of polluted sediments by the 1993 and 1996 floods for instance. These general low values likely explain why the monitoring efforts were lower on this river; LOQs were also too high to investigate the spatial and temporal evolution of the contamination of this river.

## 4.2 Spatio-temporal distribution of PCBs in each basin

To provide a spatio-temporal comparison of the PCBs contamination within each basin, ∑PCBi concentrations in sediments were compiled on the main river sections for three main time windows: < 1997, 1997–2007, 2007–2018 (Fig. 5).

A general decreasing trend of the ∑PCBi contamination is found in each river section with the highest values systematically observed before 1997 (Fig. 5). Current (2007–2018) ∑PCBi contents in sediment correspond to a

low toxicity level, usually under ecotoxicological thresholds (TEC=59.8 µg.kg$^{-1}$ cf. MacDonald et al., 2000) and under the French regulatory level N1 (80 µg.kg$^{-1}$, order of 14-07-2014) ruling the management of dredged sediments from ports and estuarine areas. These relatively low ∑PCBi accumulations in current river sediments contrast with high ∑PCBi concentrations found in related biota. For instance, the Seine estuarine mussels are inedible due to median concentrations of 330 µg.kg$^{-1}$ of PCB-153 – only one of the PCBHC (Claisse et al. 2006).

According to the OSPAR commission (2000), on the French Atlantic coast, the highest PCB concentrations in biota (mussels, oysters and fishes) next to the Loire and Garonne estuaries could be interpreted as referring to higher levels of urbanization in these rivers than along the coast in general. The highest PCB contents in surface sediments and marine biota in Spain (Eastern Biscay Bay) and in Southern England also likely indicate that French rivers provide a significant amount of contaminated sediments to these areas (OSPAR commission, 2009). In any case,

current river sediment concentrations still exceed the health-based bench-marks for freshwater fish consumption estimated at 10–27 µg.kg$^{-1}$ of PCBs in sediment (Babut et al., 2012; Lopes et al., 2012; Mourier et al., 2014).

Overall, mobile and deposited sediments showed a major increase of ∑PCBi in a downstream direction during the three studied time windows. On the Seine River, this situation is the result of a major contamination from Paris to Rouen, where the largest French urban and industrial areas are located next to the river and on its tributaries: Eure,

Yonne and Marne Rivers (Figs. 3-a, 3-b and 5). However, in 1997–2007 and in 2007–2017, ∑PCBi concentrations in fluvial sediments were higher in the Parisian stretch than in the lower section (Fig. 5-a). It might be explained





by a release of PCBs from soils and aquifers contaminated by urban and industrial activities. Indeed, a high number of accidental PCBs discharges – mainly "Pyralene oils" from electrical transformers is mentioned in this area (Fig. 3-d). This pollution may have dispersed down to the estuary where PCB contents increased in mussels (*Mytilus edulis*) from 1995 to 2006 (Tappin and Millward, 2015).


In the Rhône sediments, low to moderate ∑PCBi concentrations were generally observed before 1997 and a gradual decrease is found from 1997 to 2015 (Fig. 5-b). In the middle part of the Rhône River, the PCB pollution is likely due to emission derived from Lyon and those industrial activities from the Chemical Corridor and the Gier Valley. Concentrations on the Lower Rhône are very close to the Middle Rhône concentrations and suggest a long-distance diffusion of this pollution. Moreover, on the lower section, ∑PCBi concentrations are expected to be diluted by the high sediment load of some tributaries, such as the Durance River (1 to 2 Mt.year$^{-1}$; Poulier et al., 2019). Our hypothesis, based on the compilation of numerous data analysis, clearly contrasts with the previous assumption – based on a total of 8 cores of which only one was located on the lower section – of an exponential increase of PCB pollution in the downstream direction (Mourier et al., 2014).


The Fig. 5-c also displays very high ∑PCBi levels in the Loire River sections before 1997. In the Upper Loire, a major increase occurs in the St-Etienne-Roanne Basin which would be mainly supplied by two tributaries, i.e. the Ondaine and Furan Rivers (Fig. 3-i). This basin was historically one of the main coal extraction areas in France with a high density of urban population and industries (gas, weapons, tools…) until the 1980s. As for PAHs (Bertrand et al., 2015), the Ondaine-Furan corridor was likely the main contributor to the Upper Loire PCB contamination. In the Middle Loire River stretch, several urban and industrial zones are found (Nevers, Orléans, Tours), albeit the low quantity of data available does not allow to evaluate PCB pollution trends originating from those areas. In the Lower Loire, ∑PCBi decreased from 1997 to nowadays (Fig. 5-c). This local PCB pollution refers to urban, industrial, refining and harbour activities around the city of Nantes and the Port of St Nazaire.



The Garonne River follows a decreasing ∑PCBi trend in each section, from 1997 to 2017 (Fig. 5-d). Concentrations around Toulouse (Middle Garonne) are the highest. This result suggests a major influence of the Toulouse agglomeration on PCBi trends. Few data were available for the Pyrenees foothills before 2010 although ∑PCBi concentrations were higher compared to those lower sections at the same period. One hypothesis could be the presence of local pollution sources derived from electricity production (cf. 3.2). Moreover, the PCB pollution in the Pyrenees is also demonstrated by the local prohibition to fish eel, barber, bream, carp and catfish in Upper Garonne since 2011 (according to prefectoral orders collected by Robin des Bois, 2013). These observations of PCB contamination in sediment and fishes illustrate the continuing need to challenge the PCB pollution issue in the Garonne River (Brunet et al., 2007). Thus, current research underlines the need to acquire more precise monitoring and long-term data in several river sections that are poorly documented (e.g. Pyrenees foothills and Middle Loire, for the Garonne and Loire Rivers respectively).




## 4.3 Specific ∑PCBi fluxes and worldwide comparison

Mean annual ∑PCBi fluxes (kg.year$^{-1}$) during four decades (1977-1987, 1987-1997, 1997-2007, 2007-2017) were calculated according to the mean ∑PCBi concentrations in the lower river part and the corresponding SPM/solid fluxes during the same time windows. Then, it was normalized by respective river catchment areas to obtain specific ∑PCBi fluxes expressed in µg.m$^{-2}$.year$^{-1}$. Such process allows an estimation of annual pollutant fluxes reaching the river mouths (Babut et al., 2016; Mäkelä and Meybeck, 1996).


Using this approach at the French nationwide scale, we demonstrated that the ∑PCBi load was the highest on the Lower Rhône River, regardless of the time window considered. Indeed, specific ∑PCBi fluxes$_{Rhône}$ reached 12±3 µg.m$^{-2}$.year$^{-1}$ between 1977 and 1987 (Fig. 6). Moreover, specific ∑PCBi fluxes$_{Rhône}$ remained relatively high until 2007 (exceeding 4 µg.m$^{-2}$.year$^{-1}$), before falling to less than 1.3 µg.m$^{-2}$.year$^{-1}$ during the last decade. The Rhône could be considered as one of the main contributors to the PCB pollution in the Western Mediterranean, where shelf deposits accumulated an average of 10–30 µg.m$^{-2}$ of PCBs a year (maximum = 45–65 µg.m$^{-2}$.year$^{-1}$) from Monaco to Catalonia (Marchand et al., 1990; Salvadó et al., 2012; Tolosa et al., 1997). However, the specific ∑PCBi fluxes$_{Rhône}$ remained in the low quartile of the most polluted rivers of the world, compared to American (∑PCBi fluxes$_{Lakes Erie and Ontario}$= 0.2–11 mg.m$^{-2}$.year$^{-1}$ from 1997 to 2000: Marvin et al., 2004; ∑PCBi fluxes $_{Mississippi-Louisiana-Florida Bay}$ = 12–390 µg.m$^{-2}$.year$^{-1}$: Santschi et al., 2001) or Asian examples (e.g. ∑PCBi fluxes$_{Pearl Delta}$= 86–187 µg.m$^{-2}$.year$^{-1}$ from 1980 to 1994: Mai et al., 2005).



The Seine specific ∑PCBi fluxes amounted to ca. 8.3 µg.m$^{-2}$.year$^{-1}$ in 1977–1987, although they rapidly decreased below 1.5 µg.m$^{-2}$.year$^{-1}$ after 1987. Current specific ∑PCBi fluxes$_{Seine}$ can be estimated to ca. 0.35 µg.m$^{-2}$.year$^{-1}$



those are close to local atmospheric fluxes measured by Chevreuil et al. (2009). According to Tappin and Millward (2015), the Seine River could be considered as a major source of micropollutants to the English Channel, supplying more than half of the PCBs fluxes ($\sum$PCBi fluxes $_{English\ Channel}$ = 7.6 µg.m$^{-2}$.year$^{-1}$ in the 1980s). The Seine River also very likely contributed, together with the Thames and Rhine rivers, to the North Sea PCB pollution until 1987; however some works also show that a more local pollution, derived from small coastal rivers, apparently controls
current PCB influxes (Everaert et al., 2014; Nicolaus et al., 2015; Vandermarken et al., 2018).

When taking into account specific $\sum$PCBi fluxes, the Loire River has a PCBi load very similar to those found in the Seine River. Indeed, specific $\sum$PCBi fluxes$_{Loire}$ culminated at ca. 8.2±2 µg.m$^{-2}$.year$^{-1}$ in the 1970s and 1980s, before a sharp decrease under 2 µg.m$^{-2}$.year$^{-1}$ after 1987 (Fig. 6). A particularity of this river is that very high specific $\sum$PCBi fluxes originate from its upper section (St-Etienne-Roanne Basin). Half of the $\sum$PCBi fluxes until
2007 (59–41 kg.y$^{-1}$) were exported from the Furan/Ondaine Corridor, where high sediment loads transited (Gay et al., 2014). Finally, the Garonne River transported relatively stable amounts of $\sum$PCBi from 1977 to 1997 (2 to 1.7 µg.m$^{-2}$.year$^{-1}$), before decreasing to less than 0.5 µg.m$^{-2}$.year$^{-1}$ after 2007. Finally, $\sum$PCBi fluxes display an asymptotic behaviour (Fig. 6) which suggests the persistence of non-null background values of PCB in fluvial sediments transiting rivers in the future.

**5 Data availability**

The dataset presented in this study is freely available on the Pangea portal at: https://doi.pangaea.de/10.1594/PANGAEA.904277 (Dendievel et al., 2019).

**6 Conclusions**

In this research, we provided an original intercomparison of PCB pollution trends along four major rivers – i.e.
from source to estuary – of Europe (Seine, Rhône, Loire and Garonne Rivers). The dataset targeted the sum of the seven regulatory indicator PCBs ($\sum$PCBi) through the collection of bed and flood sediments, SPM, dredged sediments and core sediments coming from monitoring data or collected in the framework of research projects. Long-term $\sum$PCBi concentrations and fluxes were reconstructed over the last 80 years (1945–2018). The quality of the data varied according to the studied hydrosystem: the Seine and Rhône Rivers were well documented in
terms of monitoring and research data, whereas a rather low proportion of useful $\sum$PCBi data (> LOQs) was available on the Garonne and Loire Rivers. After using a correction factor on low quality data where only the highly chlorinated PCB-congeners (PCBHC) were well quantified, our results identified some major industrial and urban areas as PCBi sources, diffusing the pollution from the upper and middle river sections to the downstream areas. Two major temporal trends were found, depending on the river: (1) major and highly concentrated $\sum$PCBi releases
(up to 4 mg.kg$^{-1}$) until 1975, followed by a sharp decrease in $\sum$PCBi until today (due to the implementation of environmental regulation) occurred on the Seine and the Loire Rivers. (2) Moderate $\sum$PCBi concentrations with a longer-term diffusion until the 1990s with sporadic increases (up to 2 mg.kg$^{-1}$) due to urban or industrial uses and incidental releases (old transformers and capacitors) are found on the Rhône and Garonne Rivers. Specific $\sum$PCBi fluxes and loads since 1977 show that the Rhône provides a major quantity of $\sum$PCBi to the sea, followed by the
Seine and the Loire Rivers. In contrast, low exports from the Garonne River were found. Despite the lack of a global evaluation of PCBs delivered by rivers to the seas (Lohmann and Dachs, 2019), we highlight the important role played by French rivers in the PCB contamination of European seas, in particular the Rhône and Seine Rivers that primarily contribute to the pollution (sediments and biota) of the Western Mediterranean and the English Channel respectively. For the future, it is important to insist on the necessity to (i) improve analytical performances
for the acquisition of more reliable monitoring data on organic pollutants from river sediments and (ii) collect sediment cores at long-term accumulation sites in order to perform robust trend analyses.

**7 Supplements**

Two supplementary tables are provided with (1) the list of the sampling locations, including the geographic coordinates, the survey period and the number of samples used (supplement Table 1) and (2) hydrological and
human settings in a buffer zone of 1 km on both side of each river, including the flow rates, population distribution, land use (urban and industrial areas) and number of PCB polluted sites (supplement Table 2).





**Supplement Table 1: List of the sampling locations, chronology and number of samples accepted and corrected.**

**Supplement Table 2: Hydrological and human settings along each studied river (Garonne, Loire, Rhône, Seine Rivers).**

**Author contributions.**

A-MD and BM were in charge of collecting, formatting, cleaning and analysing data. BM was also the head of the INTERPOL project. Hydrological and land use data were acquired by A-MD and QF. AC, BM, CG, FK, HB, MDebret, MDesmet, OE, PL, SA, SV, TG, TW and YC provided core data. Spatial analysis was performed by A-MD. Statistics, spatial figures and discussion were designed by A-MD, BM and with the help of SV and TW. All
authors participated to the discussion and reviewed the draft manuscript.

**Competing interests.** The authors have no conflict of interest

**Acknowledgements and funding.** The French Agency of Biodiversity (AFB) supported the INTERPOL project in charge of INTERcomparison of sediment POLlution on the main French rivers. The INTERPOL project involves
researchers from five public research laboratories in France: the LEHNA-IPE team (UMR CNRS 5023) on the Rhône River, the EPOC team on the Garonne Basin (UMR CNRS 5805), the GEHCO lab on the Loire Basin (EA 6293), the LSCE team (UMR CEA/CNRS/UVSQ 8212) and the M2C team from Rouen (UMR CNRS 6143) on the Seine River Basin. The authors are grateful to Olivier Perceval (AFB) whose inputs greatly enhanced the project releases. Among the colleagues collaborating with the INTERPOL project, special thanks are due to Marc Babut
and Hugo Lepage for their valuable discussions. We are also grateful to the following partners for sharing data and information: Grand Port Maritime de Rouen (Haropa Ports), Grand Port maritime de Nantes, Compagnie Nationale du Rhône (CNR) and the "Robin des Bois" association.

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



# Tables

Table 1 – Sources and quality of the data. Sediment matrices are divided into two groups (column n°2): deposited sediments (cores, bed and flood deposits) and mobile particles (SPM and dredged sediments). PCBi analyses (initial) represent the total number of analyses collected from monitoring data, whereas "7 PCBi > LOQ" and "corrected ∑PCBi based on PCBHC" represent only initially well quantified samples or ∑PCBi estimated on the basis of PCBHC levels which are included on the dataset presented here. The number of stations corresponds to the sites where validated analyses (i.e. 7 PCBi > LOQ or corrected ∑PCBi) are available.

| Rivers | Matrix group | Solid matrix | Stations (n)* | Time range | PCBi Analyses (Initial) | 7 PCBi > LOQ | Corrected ∑PCBi based on PCBHC | TOC (%) | Fine Fraction (% silts & clays) | Data Sources | References and Availability |
|---|---|---|---|---|---|---|---|---|---|---|---|
| Garonne | Deposited | Bed and flood sediments | 23 | 1992-2017 | 223 | 16 | 25 | 1.3 ±1.9 | 89.7 ±5.0 | Adour-Garonne Water Agency; IFREMER (ROCCHSED); UMR CNRS 5805 EPOC | http://www.naiades.eaufrance.fr https://wwz.ifremer.fr/surval Lauzent, 2017 |
| | Deposited | Cores | 1 | 1954-2011 | 12 | - | - | | | UMR CNRS 5805 EPOC | Budzinsky, Labadie & Coynel, pers. com., Morelli et al., 2016 |
| | Mobile | SPM | 3 | 1993-2014 | 52 | 0 | 7 | 1.8 ±1.0 | - | Adour-Garonne Water Agency | http://www.naiades.eaufrance.fr |
| Loire | Deposited | Bed and flood sediments | 35 | 1994-2015 | 178 | 64 | 15 | 2.2 ±2.7 | - | Loire-Bretagne Water Agency; DREAL Auvergne-Rhône-Alpes; ONEMA; IFREMER (ROCCHSED) | http://www.naiades.eaufrance.fr http://www.rhone-mediterranee.eaufrance.fr/docs/PCB/donnees/bassin_LB/resultats-sedimentsLB_2013.xls http://www.pollutions.eaufrance.fr/pcb/resultats-xls.html; https://wwz.ifremer.fr/surval |
| | Deposited | Cores | 2 | 1973-2012 | 23 | | - | | | E.A. 6293 GeHCO | Desmet et al., 2012; Grosbois et al., pers. com. |
| | Mobile | SPM | 4 | 1993-2014 | 229 | 21 | 47 | 6.8 ±4.2 | - | Loire-Bretagne Water Agency | http://www.naiades.eaufrance.fr |
| Rhône | Deposited | Bed and flood sediments | 30 | 1995-2016 | 318 | 73 | 53 | 1.07 ±1.2 | 78.4 ±30.7 | Rhône-Méditerranée Water Agency | http://www.naiades.eaufrance.fr |
| | Deposited | Cores | 13 | 1939-2017 | 327 | | - | | | UMR CNRS 5023 LEHNA | Desmet et al., 2012; Mourier et al., 2014 |
| | Mobile | Dredged sediment | 81 | 2006-2017 | 146 | 139 | 1 | 3.4 ±2.7 | 88.1 ±22.3 | CNR | https://www.cnr.tm.fr |
| | Mobile | SPM | 2 | 2011-2016 | 209 | 13 | 15 | | | OSR | https://dx.doi.org/10.17180/OBS.OSR |
| Seine | Deposited | Bed and flood sediments | 47 | 1991-2016 | 362 | 212 | 58 | 3.5 ±3.0 | 81.7 ±9.1 | Seine-Normandie Water Agency; IFREMER (ROCCHSED); Port of Rouen | http://www.naiades.eaufrance.fr https://wwz.ifremer.fr/surval http://www.haropaports.com/fr/rouen |
| | Deposited | Cores | 3 | 1945-2015 | 222 | | - | | | UMR CNRS 8212 LSCE; UMR CNRS 6143 M2C | Boust et al., 2011; Gardes et al., submitted; Lorgeoux et al., 2016 |
| | Mobile | Dredged sediment | 5 | 1992-2018 | 88 | 65 | 8 | 1.0 ±0.9 | - | Port of Rouen | http://www.haropaports.com/fr/rouen |





**Table 2 – River characteristics. Lengths are given following the SANDRE baseline datasets (http://id.eaufrance.fr/). *For the Rhône River, the length is representative of its French course. **Watershed surfaces are given upstream of the gauging stations where specific fluxes where estimated. The water-engineering column refers to groups defined in the text. SPM – Suspended Particulate Matters – discharges were based on existing literature (Moatar et al., 2006; Copard et al., 2018; Descy et al., 2009; Olivier et al., 2009).**


| River | Length (km) | Altitudinal range (m) | Watershed (km²)** | Flow rate (m³.s⁻¹) | SPM discharge (Mt.year⁻¹) | Cumulated Population (Minhab.) | Main Towns (downstream direction) | Water-engineering |
|---|---|---|---|---|---|---|---|---|
| Seine | 775 | 445–0 | 65366 | 600 | 0.2–0.6 | 20.1 | Troyes, Paris Megacity, Rouen, Le Havre | Group 1 |
| Rhône | 545* | 347–0 | 89011 | 1700 | 4–8 | 12.2 | Geneva, Lyon, Valence, Avignon, Arles | Group 1 |
| Loire | 1006 | 1551–0 | 110726 | 870 | 0.4–0.7 | 14.3 | St Etienne, Nevers, Orléans, Tours, Angers, Nantes | Group 2 |
| Garonne | 529 | 1840–0 | 51257 | 650 | 0.9–3 | 5.9 | Toulouse, Agen, Bordeaux | Group 2 |



**Earth System Science Data Discussions**
**Figures**

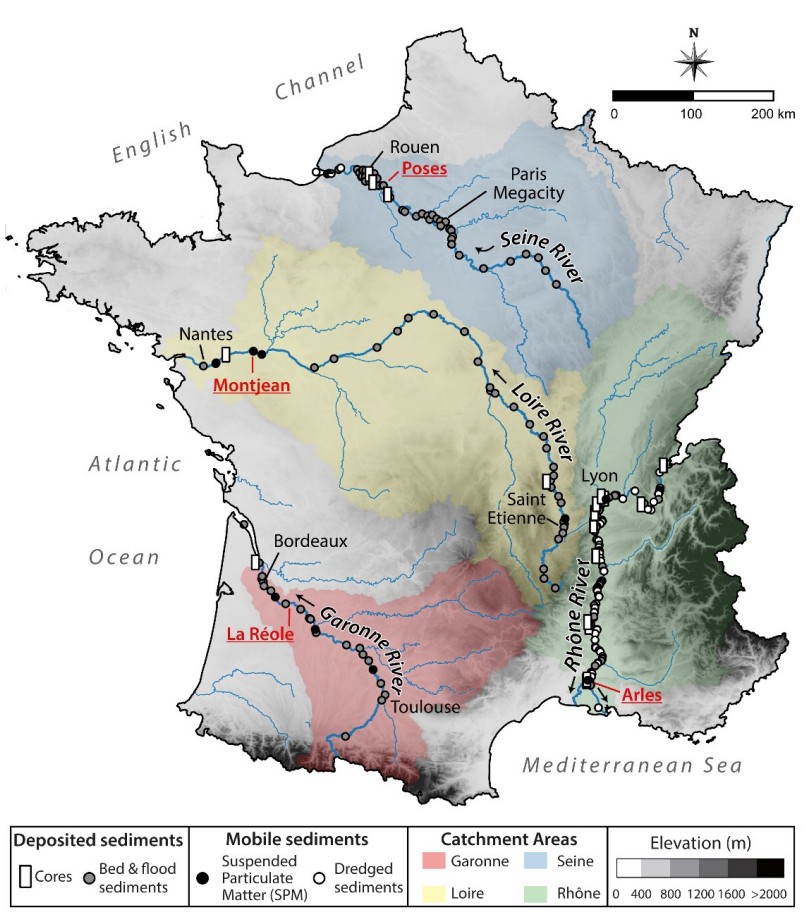

**Fig. 1 – Location of sampling stations along the four main French Rivers (Garonne, Loire, Rhône, Seine). Main cities are reported in black whereas locations mentioned in red and underlined correspond to the four stations where specific fluxes are estimated (cf. 4.3).**



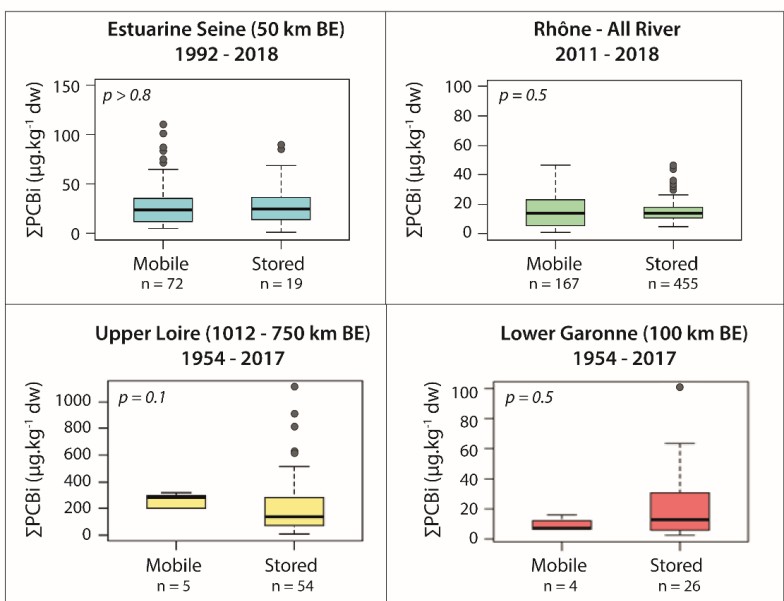

**Fig. 2 – Comparison of the ∑PCBi distribution within the different sediment matrices of the database. Distances are expressed in kilometres before the estuary (BE) of each river. Graphics and related Wilcoxon tests were performed with *R* (v. 3.5.1, R Core Team, 2018).**



**Fig. 3 – Main physical settings, population, land use and PCB concentrations along each river: a-d) Seine, e-h) Rhône, i-l) Loire, m-p) Garonne. Numbers in boxes (a), (e), (i) and (m) represent the main dams (Da), harbours (Hb), channels**





**and weirs linked to hydroelectric (HP) or nuclear power plants (NP). ∑PCBi concentrations in sediment represented in boxes (c), (g), (k) and (o) are expressed as µg.kg$^{-1}$ (dry weight) and represented with a log scale. Boxes (d), (h), (l), (p) showed the number of sites with confirmed (transformers or capacitors leakage mainly) or suspected (e.g. wood, paper,**

**ink additives, flammable fluid storage) PCB contamination. Box (a):** 1) Burgundian ponds, 2) Trojan weirs and mills, 3) Haute-Seine channels and Nogent NP, 4) Ablon Da, 5) Suresmes Da, 6) Gennevilliers Hb, 7) Chatou Da, 8) Bougival Treatment Plant, 9) Andrésy Da, 10) Paris Hb, 11) Méricourt Da, 12) Port Mort Da, 13) Poses Da, 14) Rouen Hb. **Box (e):** 1) Verbois Da in Switzerland (Switzld.), 2) Pougny Da, 3) Génissiat Da, 4) Bregnier-Cordon Da, 5) Villebois Da, 6) Grand-Large Da, 7) E. Herriot Hb, 8) St-Alban NP and St-Pierre-Bœuf Da, 9) Arras-sur-Rhône Da, 10) Charmes-sur-Rhône Da, 11) Cruas

NP and Rochemaure Da, 12) Donzère Da, HP and NP, 13) Caderousse Da, 14) Roquemaure Da, 15) Beaucaire Da. **Box (i):** 1) La Palisse Da, 2) Loire Gorges HP, 3) Grangent Da, 4) Villerest Dam, 5) Decize HP, 6) Dampierre NP, 7) St-Laurent-les-Eaux NP, 8) Chinon NP, 9) Groynes "Epis de Loire", 10) St-Nazaire Hb. **Box (m):** 1) Pont-du-Roi Da, 2) Camon, St Sernin and St Martory HP, 3) St Vidian Da, 4) Labrioulette Da, 5) Mancies Da, 6) Ramier and Bazacle HP, 7) Malause Da, 8) Golfech NP, 9) Side structures, 10) Bordeaux Hb.

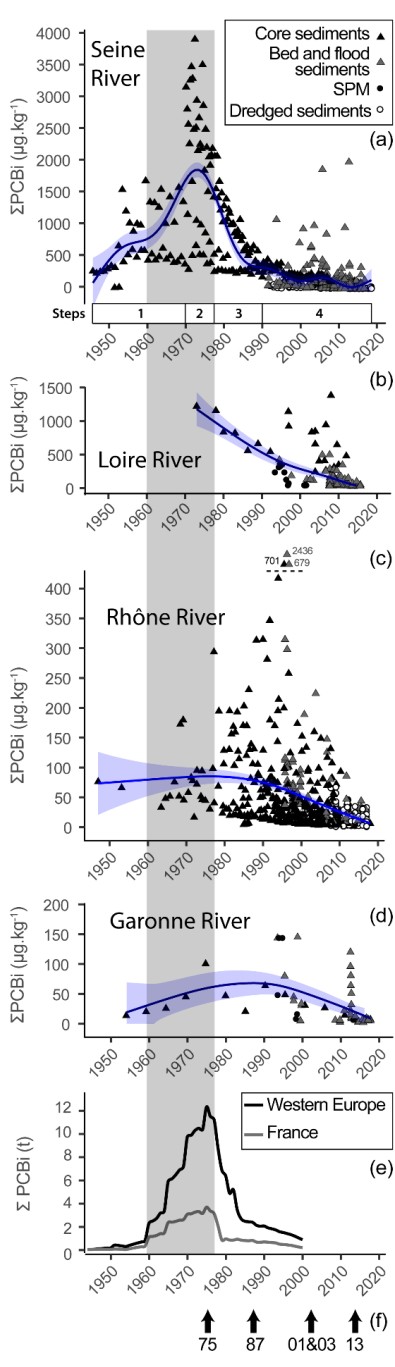


**Fig. 4 – Estimated trends of ∑PCBi in fluvial sediments since 1945 on the 4 basins: (a) Seine, (b) Loire, (c) Rhône, (d) Garonne. The blue curve represents the gam modelling and the pale blue area its confidence interval. Steps 1, 2, 3, and 4 refer to the Seine trend cited in the text. (e) Comparison with global estimated emissions of ∑PCBi in tons for France and Western Europe (after Breivik et al., 2002a, 2007). (f) Main regulations on PCBs. 75: French restriction to closed**
**devices (order of 08-07-1975); 87: prohibition of production, sale and purchase of devices using PCBs > 500 mg.kg$^{-1}$ (decree 87-59); 01&03: removal of devices using PCBs > 500 mg.kg$^{-1}$ (decree 2001-63 and order of 26-02-2003); 13: disposal of devices using PCBs > 50 mg.kg$^{-1}$ (decree 2013-301). The grey bar underlines the main period of PCBs production worldwide.**

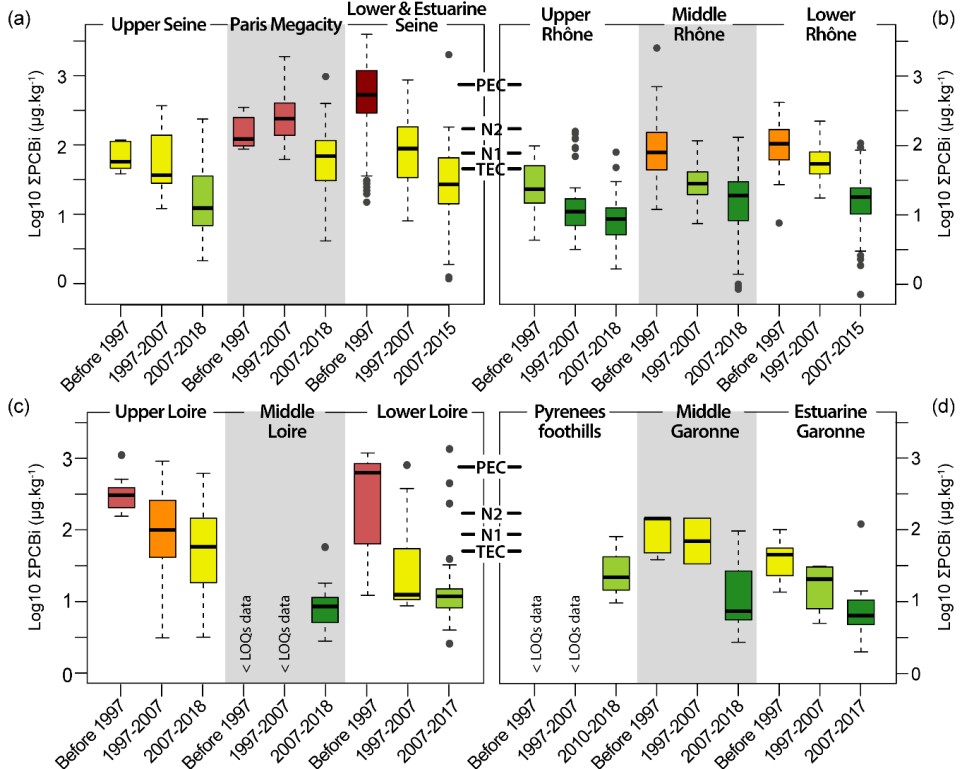

**Fig. 5 – Detailed spatio-temporal distribution of ∑PCBi (log10 scale). (a) Seine River, (b) Rhône River, (c) Loire River, (d) Garonne River. Colours describe distinct pollution levels in sediments, which are ranked from green (lowest levels) to red (highest levels). Similar colours indicate comparable distributions of ∑PCBi between the successive river sections. TEC, PEC, N1 and N2 symbolized the main levels of PCBs pollution in river and estuarine sediments. TEC (Threshold Effect Concentration) and PEC (Probable Effect Concentration) are estimated to 59.8 µg.kg$^{-1}$ and to 676 µg.kg$^{-1}$ for total PCBs, respectively (MacDonald et al., 2000). French regulatory levels N1 and N2 refer respectively to 80 µg.kg$^{-1}$ and 160 µg.kg$^{-1}$ of ∑PCBi (order of 17/07/2014).**

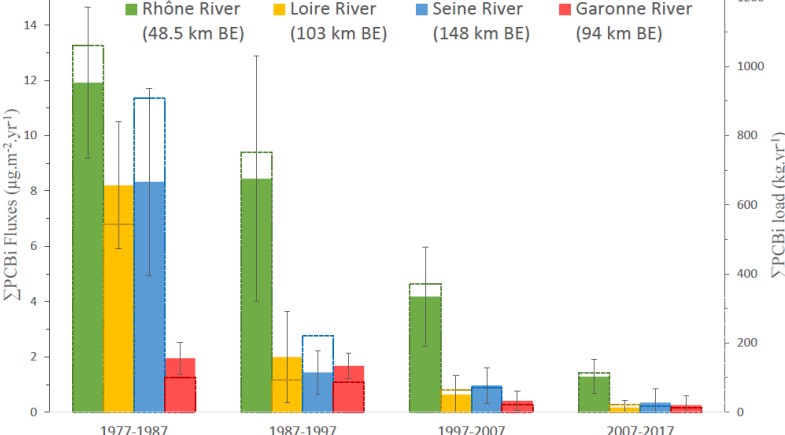

**Fig. 6 – Comparison of specific ∑PCBi fluxes (coloured rectangles, µg.m$^{-2}$.year$^{-1}$) and mean annual loads (dashed rectangles, kg.year$^{-1}$) estimated at the lower station upstream of the tidal influence on the main French Rivers since 1977 (see Fig. 1 for details). The distance of each station to the sea is expressed as kilometres before estuary or delta (km BE).**