# Peer review of "Spatio-temporal assessment of the PCB sediment contamination in four major French river corridors (1945-2018)"

_Earth System Science Data, 2019_

## Referee Comment (RC1) · Marc Babut (Referee) · 3 Jan 2020

Numbers in brackets refer to the line number.

General comments This study attempts summarizing sediment contamination data for a long period of time, and at a large spatial scale, from very different sources, which is quite challenging. Some findings, such as the relative importance of the Rhone and Seine Rivers for the Mediterranean Sea and the English Channel, are not that new, but the broad perspective of the study is certainly interesting. In my view, this article is worth publishing after revision (see specific comments below).

Specific comments I. Data handling and management: indeed, as stressed by the authors, quality control is a major challenge in this kind of study. I am afraid that this challenge was not addressed properly, on several aspects: - (111) compiling data from different sources (databases) was indeed a necessity, raising accordingly the possibility of having duplicates in the final datasets. There is in particular a concern regarding item b (database from the National Action Plan): these data are presumably also recorded in the Naïades database, as they were originally extracted from Water agencies monitoring records. Did you check for duplicates? - (156) LOQ for the sum of congeners is not equivalent to the sum of individual LOQs, but rather to the LOQ for one congener, or the highest LOQ among all congeners if they have different LOQ values. If the concentrations of six congeners were below their respective LOQs, and the concentration of the 7th would be > LOQ, ÆľPCBi would be at least above the LOQ of this 7th congener, while still below the sum of LOQs. This seems weird . . . - Two kinds of ÆľPCBi are considered as valid and selected by the authors (157-166): (i) when all seven congeners > LOQs (simple case), and (ii) when only highly chlorinated congeners are >LOQ, ÆľPCBi is estimated according to equation (1), where censored values are replaced by an estimate derived from the mean proportion of these highly chlorinated congeners in uncensored samples. This cannot at all maximize the robustness, as claimed by the authors, it is the opposite (Helsel, 2006). There are better much options for summing non-detects, as explained in (Helsel, 2010). - A brief description of the characteristics of the dataset would be useful. - (219) The Wilcoxon paired test seems sufficient for comparing both groups of matrices; graphical comparison is not a robust approach. II. There are too many figures, and not all are useful (e.g. Figure 2, Figure 4 could be skipped). Furthermore, Figure 3 is almost impossible to read and understand as a whole and does not help the reader to follow the arguments. Scales are not systematically comparable, so the benefit of having all this information summarized on the figure is hampered by the precautions that has to be made when looking at the figure. I suggest to alleviate the figure, remove some parts of it and put it in annex, allowing to enlarge the scales of the remaining parts. III. Section 3.1 –

typology: the different kinds of sediment data were put together in two groups, either "deposited" or "mobile". This seems a good idea, for the sake of statistical robustness. Nevertheless, the rationale behind the assignment of sub-categories into one group or the other could be made clearer. More specifically, a probably significant amount of dredged sediments in the Rhone River originate from more or less recent deposits in critical locations of the waterways (e.g. locks). Should these materials be considered as mobile instead of deposited? It seems somewhat surprising that the contamination level does not differ between the two groups; how much the grouping rationale does influence this result? IV. Section 3.4: - (295-300) "Part of this increase might be due to the Eure river"; the figure 3 (c and d) is somewhat difficult to read, due to the respective sizes, but it seems that most of the PCB contaminated sites (3d) are located well upstream from the confluence of the Seine river with this tributary. Are there some specific arguments justifying this statement? Why not assigning also elevated PCB concentrations in the estuary to long term fine sediment particle transport from the Paris area? - (305-306) decreasing trend of ÆľPCBi in the lower Rhone River section: does this statement result from a statistical test? This trend seems doubtful, considering the median concentration in the last two river sections. (respectively 32 $\pm$ 24 $\mu$g kg-1 and 24 $\pm$ 18 $\mu$g kg-1). - (313-320) ". . . PCB contaminated sites are reported along the rivers . . ." sounds somewhat ambiguous, according to the current title which refers to catchments. Among the sources that are mentioned, PCB production should not be omitted; one of the two historical French production sites is located in the Rhone River basin. This remark could also be accounted for in the discussion (section 4.1, 355-372) V. Sections 4.1 and 4.2 are quite redundant; moreover, Fig. 5 seems more illustrative than Fig. 4 (which could be deleted or skipped to the annex).

Technical issues 1. A thorough language edition is recommended. Among others, repeated mistake "upstream (or downstream) from ..."; inappropriate use of articles, and so on. 2. Title: suggestion "Spatio-temporal trend assessment of PCB contamination in four major French rivers (1945-2018)" 3. When expressing units (either concentrations or flows and so on), the dot should be avoided: for instance, "ng g-1" is correct, not

"ng.g-1" (https://www.bipm.org/en/publications/si-brochure/ ) 4. Abstract: (34) "highest" rather than "biggest"; (37) "provide a major supply" is unclear. 5. Keywords: the term "pollution trajectories" seems useless as keyword (not found as such in bibliographic databases). It is not clear whether this term refer to the temporal trends of PCB concentrations, or to their downward transport. 6. Introduction: (45) confusing chronology: "a concern as early as in the 1970s . . . PCBs were then used as heat transfer fluid". The PCB use as heat transfer fluid was a major use well before the 70s. (65) "compared sediment and biota concentrations" was this comparison the actual purpose of these studies? Be more specific. 7. (101-102): misleading and erroneous term "regulatory water agencies". Suggestion to rephrase ". . . and monitored for regulatory purposes". 8. (157) PCB-28; (160) apparently a word missing: "we corrected the sample results where only high-chlorinated congeners . . . were > LOQs" (without this word, the sentence does not make sense). 9. (219) word missing: graphical comparison (?); please refer also to my comment in the previous section. 10. (330) Why "theoretical production trend"? PCB production along several decades is not a theory, it is a fact. I suggest to replace this inappropriate wording by "estimated production inventory". 11. (400-402) this writing sounds weird: "current river sediment concentrations exceed health-based benchmarks for freshwater fish consumption". Moreover, the so-called fish consumption benchmark was updated after the cited publications (Vigreux-Besret et al., 2015) 12. Figure 6 legend: improper term "dashed rectangles". 13. Table 1: wrong spelling "Budzinski".

References cited Helsel DR (2006). Fabricating data: How substituting values for non-detects can ruin results, and what can be done about it. Chemosphere, 65 : 2434-2439. Helsel DR (2010). Summing nondetects: Incorporating low-level contaminants in risk assessment. Integr Environ Assess Manag, 6 : 361-366. Vigreux-Besret C, Rivière G, Feidt C, Amiard J-C, Babut M, Badot P-M, Blanchemanche S, Camel V, Le Bizec B, Narbonne J-F, Roudot A, Vernoux J-P, Volatier J-L, Mahé A, Desvignes V (2015). Consommation de poissons d'eau douce et PCB : aspects réglementaires, méthodologiques et sanitaires. Avis de l'Anses - Rapport d'expertise collective, Agence

Nationale de Sécurité Sanitaire, Alimentation, Environnement, Travail

---

## Referee Comment (RC2) · Anonymous Referee #2 · 11 Jan 2020

Line 190 – 205: Clarify the methodology for calculating PCBs fluxes and loads. Are PCBs fluxes estimated on the basis of cores chronology and estimated sediment accumulation rates/mass accumulation rates? Line 290-320: Are there differences in the mean concentrations of PCBs in the specific time periods moving downstream? Line 330 – 340: Are there more recent to threshold effect concentrations to compare with? (i.e. predicted non-effect concentration (PNEC) from sediment bioassays). Are there PCBs reference background levels and assessment levels for the Mediterranean Sea and/or from OSPAR to compare with?

[Figure]

2019.

---

## Author Comment (AC1) · 27 Feb 2020

Many thanks to RC1 for his comments and critiques, which allowed to improve the quality of the manuscript. Based on your comments, major changes will be made to clarify and better explain the methodology of data collection and the handling of values below the quantification levels (in parts 2.2 and 2.4). The title of the paper will be modified as follows: "Spatio-temporal assessment of the PCB sediment contamination in four major French river corridors (1945-2018)". We will revise the design of the study regarding the presentation of socio-environmental data by summarizing or improving the elements of the new version of fig. 2 - spatial comparison of PCBs concentrations

with geographical and socio-environmental drivers (former fig. 3 of the draft; see figure 1 attached to this answer). We will prepare a new version of fig. 5 – PCBi specific fluxes and load histograms (former fig. 6 of the draft; see figure 2 attached to this answer). In parallel, we will review part 3 (results) of the draft. The former fig. 1 will be converted into a supplementary figure (suppl. Fig. 1 - see figure 3 attached). We also propose to develop and add more references about the interpretations of the pollution sources on the Lower Seine River. We will revise the text dealing with the interpretations of PCBs production and emission along the Rhône River. Then, we will rewrite the second part of the discussion (part 4.2) to include a comparison between the contamination of the sediment by PCBs and the implications for biota from the source to the estuary on the studied rivers.

[Figure]

**Fig. 1.**

[Figure]

**Fig. 2.**

[Figure]

**Fig. 3.**

---

## Author Comment (AC2) · 27 Feb 2020

We are grateful to RC2 for his constructive remarks that helped us improving some methodology aspects and developing the discussion. To address your questions, we will clarify the methodology employed for the calculation of PCBi fluxes (based on monitored flow rates and suspended matter transported by each river). We will also expand the text with the interpretations of the spatial and temporal trends of the PCB dispersion from upper to lower catchment parts, especially in the Seine and Rhône Rivers. We also add as supplementary data a comparison of two maps displaying the evolution of maximal concentrations in PCBs in fluvial sediments from 1945 to the 2000-2018 pe-

riod along each river (see fig. 1 attached to this reply). Finally, following your advice, in the second part of the discussion, we will include other threshold effect concentrations such as the OSPAR EAC (Environmental Assessment Criteria) to discuss the spatial and temporal relationships between PCBis in sediment and their accumulation in the biota.

———————————————————

[Figure]
* * *
[Figure]

**Fig. 1.**

---

## Author Response (AR1)

Revision – Detailed answers to the reviewers

We are grateful to the two reviewers for their careful reading, their insightful comments and suggestions which helped us to improve the quality of our manuscript. Please find our detailed answers to your questions and the related changes (line quotation) in the following table.

| Comment from the reviewer | Answer | Correction done |
|---|---|---|
| **RC1 General comment** | | |
| This study attempts summarizing sediment contamination data for a long period of time, and at a large spatial scale, from very different sources, which is quite challenging. Some findings, such as the relative importance of the Rhone and Seine Rivers for the Mediterranean Sea and the English Channel, are not that new, but the broad perspective of the study is certainly interesting. In my view, this article is worth publishing after revision (see specific comments below). | Thank you for this encouraging general comment. | |
| **RC1 Specific Issues** | | |
| Data handling and management, question on line 111: the PCB National Action Plan data are presumably also recorded in the Naïades database, as they were originally extracted from Water agencies monitoring records. Did you check for duplicates? | The original data and sources were carefully checked, especially to avoid duplicates. It is correct that some data were mentioned by several databases such as *Naiades* (accessed October 15, 2018). In this case we have removed the duplicates. However, these background checks also demonstrated that some data were not included in *Naiades*. For instance, on the Loire River, we added 8 analyses performed from 2008 to 2010 between PK 820 and 343 (from Veauchette to Meung-sur-Loire). | Based on this reply, we have added a short precision to the text: "and checked to avoid potential duplicates" (lines 111-112) |
| Question on line 156: LOQ for the sum of congeners is not equivalent to the sum of individual LOQs, but rather to the LOQ for one congener, or the highest LOQ among all congeners if they have different LOQ values. If the concentrations of six congeners were below their respective LOQs, and the concentration of the 7th would be > LOQ, ∑PCBi would be at least above the LOQ of this 7th congener, while still below the sum of LOQs. This seems weird... | We agree with your comment and have rewritten to clarify the text because this is not what we meant. LOQs are heterogeneous and the LOQ for the sum of congeners was cited for information purpose.
The new version of the description insists on the individual LOQs which were taken into account to build our dataset (because the PCB-congener concentration is the key information presented in the dataset). | Changes were done and an example (from the Garonne River) was added from line 155 to 158 |
| Questions on lines 157-166: the correction method used by the authors cannot maximize the robustness (Helsel, 2006). | Based on the dominance of undetected values (related to too high and varying LOQs) in several | To address this issue we added |

| | | |
|---|---|---|
| There are better much options for summing non-detects, as explained in (Helsel, 2010). - A brief description of the characteristics of the dataset would be useful | river sections during more than 10 years (e.g. medium sections of the Garonne and the Loire Rivers), we chose to estimate PCB levels based on measured PCB-HC concentrations and on a mean-imputation of PCB-LC, adapted to each river section, and based on the samples in which all PCB congeners were detected. | precisions about this method and a brief description of the dataset in the updated text (lines 161-167) |
| Question on line 219: The Wilcoxon paired test seems sufficient for comparing both groups of matrices; graphical comparison is not a robust approach | We agree, maybe this sentence was not well-phrased because the comparison was not based on the graph: Wilcoxon paired tests were performed and p-values (between 0.1 and 0.8) indicate that the distribution was not significantly different between the studied matrices. | We clarified the text (l. 224-226) and moved the former fig. 2 (boxplots) in supplement |
| There are too many figures, and not all are useful (e.g. Figure 2, Figure 4 could be skipped). | We have moved Figure 2 to the supplementary data. However, we think Figure 4 is useful for properly describing the time-patterns of the PCB pollution in each river. In addition, we have divided figure 5 (histograms) into two plots to facilitate the reading. | |
| RC1 suggests to alleviate the figure 3: remove some parts of it and put it in annex, allowing to enlarge the scales of the remaining parts | In this revised version of the manuscript, we propose a simplified version of this figure (now numbered fig. 2). We have deleted longitudinal profiles because this data was little used. We have re-organised the socio-environmental boxes of Fig. 2, and proposed a new representation of polluted sites only based on the PCB contaminated sites with a cumulative curve. | |
| Section 3.1 – remark on the typology: a probably significant amount of dredged sediments in the Rhone River originate from more or less recent deposits in critical locations of the waterways (e.g. locks). Should these materials be considered as mobile instead of deposited? | We agree with this statement, dredged sediments are already considered as mobile sediments in our analysis (as written on line 218) | |
| It seems somewhat surprising that the contamination level does not differ between the two groups; how much the grouping does influence this result? | According to the data, the absence of significant difference is due to comparable pollutant concentrations among the deposits. The high number of data probably smoothed the results, while the influence of the sediment parameters (TOC, grain size) is difficult to estimate because, contrary to the sediment cores, it was not systematically measured on the analysed samples. | |
| Section 3.4, lines 295-300: "Part of this increase might be due to the Eure River": it does not fit with the PCB contaminated | Indeed, the sites at the connection between the Seine River and its tributary (Eure River) are highly | We have added more precisions |

| | | |
|---|---|---|
| sites. Are there some specific arguments justifying this statement? Why not assigning also elevated PCB concentrations in the estuary to long term fine sediment particle transport from the Paris area? | contaminated (up to 1.5 mg/kg; see black triangles at 133 km upstream of the estuary on the revised fig. 2d). Nevertheless, we agree with the reviewer: it is likely that a significant part of the pollution originates from upstream-sites. | and references to the revised text (l. 298-301). |
| Lines 305-306: decreasing trend of ∑PCBi in the lower Rhone River section: does this statement result from a statistical test? This trend seems doubtful, considering the median concentration in the last two river sections (respectively 32±24 g kg-1 and 24±18 g kg-1) | This trend was established according to the gam model, so that we are confident about reporting it.
To address the reviewer comment, we have modified the text as follows: "Then, according to the gam modelling, the Lower Rhône presents a slight decrease of the PCB contamination (median: 24±18 $\mu$g kg$^{-1}$)" (l. 306) | |
| Lines 313-320: "PCB contaminated sites are reported along the rivers" sounds somewhat ambiguous, according to the current title which refers to catchments. Among the sources that are mentioned, PCB production should not be omitted; one of the two historical French production sites is located in the Rhone River basin. This remark could also be accounted for in the discussion (section 4.1, 355-372) | We totally agree and have revised the title of the paper (river corridors) accordingly.
We have also added some details in the discussion regarding the production of PCBs (lines 407-409 for instance). | |
| Sections 4.1 and 4.2 are quite redundant; moreover, Fig. 5 seems more illustrative than Fig. 4 (which could be deleted or skipped to the annex). | The reviewer is right, so we have modified sections 4.1 and 4.2 to better underline their own specificities. In part 4.1, we have insisted on the comparison of PCB temporal trends between the rivers by providing a more detailed interpretation of the temporal pattern obtained for the Rhône River for instance.
We kept Fig. 4 to illustrate our purpose and hypotheses. This figure underlines different temporal trends between the rivers, probably due to the variability of processes, emissions and resilience. Fig. 4 also supports the hypothesis that the Loire River has a temporal behaviour close to the Seine's trend. Then, we have reorganized section 4.2 to better integrate the spatio-temporal aspects of the pollution in rivers and changed the title of the section as follows: "Spatio-temporal distribution of the PCB contamination and implications for biota" to discuss the diffusion of the pollution towards the sea and the results outlined by fauna analyses. | |
| **RC1 Technical Issues** | | |
| A thorough language edition is recommended. Among others, repeated mistake "upstream (or downstream) from …"; inappropriate use of articles, and so on | Ok, we have corrected | |
| Title: suggestion "Spatio-temporal trend assessment of PCB contamination in four major French rivers (1945-2018)" | As explained in a previous remark, we kept "sediment" and also added "corridors" at the end of the title | |

| | |
|---|---|
| When expressing units (either concentrations or flows and so on), the dot should be avoided: for instance, "ng g-1" is correct, not "ng.g-1" (https://www.bipm.org/en/publications/si-brochure/ ) | As suggested by the reviewer, we have corrected the text and the figures |
| Abstract: (line 34) "highest" rather than "biggest" | Ok, we have changed biggest for uppermost |
| line 37: "provide a major supply" is unclear | To address this concern, we propose now "contribute significantly" |
| Keywords: the term "pollution trajectories" seems useless as keyword (not found as such in bibliographic databases). It is not clear whether this term refer to the temporal trends of PCB concentrations, or to their downward transport. | Ok, we propose "historical pollution trends" instead of "pollution trajectories" |
| Introduction: (line 45) confusing chronology: "a concern as early as in the 1970s: PCBs were then used as heat transfer fluid". The PCB use as heat transfer fluid was a major use well before the 70s. | It was not what we meant, we have therefore deleted the term "then" |
| Line 65: "compared sediment and biota concentrations" was this comparison the actual purpose of these studies? Be more specific. | The goals of these papers were slightly different. To find a consensus, we have rewritten the sentence as follows "focused on both sediment and biota in order to assess the relationships between particle pollution and accumulation in species." |
| Lines 101-102: Suggestion to rephrase "regulatory water agencies" in " and monitored for regulatory purposes". | Ok, we have corrected and changed "WBA" for "WA" abbreviations for Water Agencies in the text |
| Line 157: PCB-28. | Ok, we have corrected |
| Line 160: apparently a word missing: "we corrected the sample results where only high-chlorinated congeners…were > LOQs" (without this word, the sentence does not make sense). | This part has been entirely rewritten (lines 161-167). |
| Line 219: word missing: graphical comparison (?); please refer also to my comment in the previous section. | This expression was deleted according to your previous comment. |
| Line 330: Suggestion to replace "theoretical production trend" by "estimated production inventory". | Ok, we agree. |
| Lines 400-402: this writing sounds weird: "current river sediment concentrations exceed health-based benchmarks for freshwater fish consumption". Moreover, the so-called fish consumption benchmark was updated after the cited publications (Vigreux-Besret et al., 2015) | Ok, we have modified this sentence in order to integrate this work |

| | | |
|---|---|---|
| Figure 6 legend: improper term "dashed rectangles". | In the end, we chose to detail the two plots of this figure (numbered now Fig. 5) in order to facilitate the reading and to homogenise the keys. | |
| Table 1: wrong spelling "Budzinski" | Ok, we have corrected | |

**RC2 Comments**

| | | |
|---|---|---|
| Lines 190 – 205: Clarify the methodology for calculating PCBs fluxes and loads. Are PCBs fluxes estimated on the basis of cores chronology and estimated sediment accumulation / mass accumulation rates? | Fluxes were based on monitoring flow rates and suspended matter (SPM) fluxes delivered to the sea (source: http://www.hydro.eaufrance.fr), rather than on sediment/mass accumulation rates. Indeed, it was more relevant to estimate transiting river fluxes because the relationships between deposited and transiting flux is still difficult to quantify. Then, we used the median concentration of PCBs over 10 year-periods to obtain annual PCBi loads and specific fluxes. | Based on your comment, we added some precisions at lines 203–206. |
| Line 290-320: Are there differences in the mean concentrations of PCBs in the specific time periods moving downstream? | Actually, these differences are discussed later in the revised manuscript (part 4.2). PCBi trends in the stretch Paris-Rouen (Seine River) and from Lyon to Arles (Rhône River) certainly imply a diffuse emission or dilution in the downstream direction (see Figs. 3 & 4). Even if an upstream-downstream gradient could be evidenced, it is difficult to estimate the quantity of PCBs moving downstream (speed and distance) during the specific periods due to diverse hydrological settings. | We have added more precisions in the results (l. 295-301) and in the discussion to address your remark (lines 405-412). |
| Line 330 – 340: Are there more recent threshold effect concentrations to compare with? (i.e. predicted non-effect concentration (PNEC) from sediment bioassays). Are there PCBs reference background levels and assessment levels for the Mediterranean Sea and/or from OSPAR to compare with? | Following the reviewer comment, we have included additional information (lines 390-403 and 451-452) about the OSPAR EAC limit (Environmental Assessment Criteria = 67.9 μg kg$^{-1}$) which is between the TEC and the French N1 level. Currently, EACs are still frequently exceeded in the Seine Estuary and, for some PCB-congeners, in the Loire estuary. Regarding Probable No Effect Concentrations (PNEC) in sediments, a PNEC for total PCBs was estimated in an INERIS/IOW report of 2009, i.e. 0.4 μg kg$^{-1}$; however such value could be considered as excessively low according to Cachot et al., 2006 (*Evidence of genotoxicity related to high PAH content of sediments in the upper part of the Seine estuary (Normandy, France). Aquatic Toxicology* 79: 257–267) | |